

# Intelligent search system for resume and labor law

Hien Nguyen[1,2,*], Vuong Pham[1,3,4,*], Hung Q. Ngo[5], Anh Huynh[1,2], Binh Nguyen[1,3] and José Machado[6]

[1] Vietnam National University, Ho Chi Minh, Vietnam
[2] University of Information Technology, Ho Chi Minh, Vietnam
[3] Faculty of Mathematics and Computer Science, University of Science, Ho Chi Minh, Vietnam
[4] Institute of Data Science and Artificial Intelligence, Sai Gon University, Ho Chi Minh, Vietnam
[5] Technological University Dublin, Dublin, Ireland
[6] Centro ALGORITMI/LASI, University of Minho, Braga, Portugal
[*] These authors contributed equally to this work.

Corresponding authors
Hien Nguyen, hiennd@uit.edu.vn
Vuong Pham,
vuong.pham@sgu.edu.vn

## ABSTRACT

Labor and employment are important issues in social life. The demand for online job searching and searching for labor regulations in legal documents, particularly regarding the policy for unemployment benefits, is essential. Nowadays, each function has some programs for its working. However, there is no program that combines both functions. In practice, when users seek a job, they may be unemployed or want to transfer to another work. Thus, they are required to search for regulations about unemployment insurance policies and related information, as well as regulations about workers working smoothly and following labor law. Ontology is a useful technique for representing areas of practical knowledge. This article proposes an ontology-based method for solving labor and employment-related problems. First, we construct an ontology of job skills to match *curriculum vitae* (CV) and job descriptions (JD). In addition, an ontology for representing labor law documents is proposed to aid users in their search for legal labor law regulations. These ontologies are combined to construct the knowledge base of a job-searching and labor law-searching system. In addition, this integrated ontology is used to study several issues involving the matching of CVs and JDs and the search for labor law issues. A system for intelligent resume searching in information technology is developed using the proposed method. This system also incorporates queries pertaining to Vietnamese labor law policies regarding unemployment and healthcare benefits. The experimental results demonstrate that the method designed to assist job seekers and users searching for legal labor documents is effective.

## INTRODUCTION

When working in human resources, recruiters must find suitable candidates for recruitment based on their experience. It can be difficult to evaluate the fitness of a candidate for a job description based solely on their curriculum vitae (CV). Similarly, candidates may miss out on job opportunities that match their skills. Therefore, an intelligent resume screening

system can assist recruiters and candidates in finding suitable results more quickly and efficiently for their respective needs (*Faliagka et al., 2014*). An effective job search engine helps users quickly find new opportunities to enhance their careers and promote household financial stability. Through online job searching, users can have many chances to find jobs suitable for their skills. This system allows individuals to identify and apply their skills in areas most needed. This optimal use of skills benefits employers, leading to increased productivity and innovation.

The challenge of job matching between CVs and JDs calls for the employment of numerous machine learning and natural language processing (NLP) techniques. The task is specifically defined as follows: Given a set of job descriptions (JDs) and a set of candidate CVs, the problems are the ranking of CVs for a given JD and the ranking of JDs for a given CV (*Cannella-Malone, Bumpus & Sun, 2020*; *Faliagka et al., 2014*). In addition, the requirement for searching legal regulations of labor is necessary for candidates when they seek jobs (*NavigoGroup, 2021*). Labor regulations are designed to protect the rights and well-being of workers. By searching and understanding these regulations, both employers and employees can ensure that working conditions are safe, fair, and in accordance with established legal standards. This helps prevent exploitation and promotes a healthy work environment. Employers must adhere to labor laws and regulations to ensure legal compliance. Understanding and following these regulations helps businesses avoid legal repercussions, fines, or other penalties. This, in turn, contributes to the overall stability and reputation of the business. Thus, a resume screening system needs to combine the ability to search labor law to help users retrieve useful knowledge (*Breuker, Valente & Winkels, 2004*). This helps the built system support candidates better.

Nowadays, *Greenstein (2022)* studies of law and IT are inherently contradictory because technology helps to quickly embrace concepts in legal documents. Several problems related to CV and JD matching are proposed and solved, such as data collection, skill extraction, and skill graph generation (*Phan et al., 2021*). Moreover, ontology-based approaches effectively represent and search data based on its semantics in many domains (*Do, Nguyen & Hoang, 2020*; *Dudáš et al., 2018*). *Stojanovic et al. (2004)* defined an ontology as a formal, shared conceptualization of a particular domain of interest. Ontology defines a set of representational primitives with which to model a domain of knowledge or discourse. The representational primitives are typically concepts, attributes, and relations among concepts (*Do, Nguyen & Selamat, 2018*; *Fawei et al., 2019*). Thus, such studies form a solid foundation for research in the cross-discipline field of legal and AI.

In this article, we propose a new ontology-based method for solving problems related to labor and jobs. This study provides solutions for several problems searching for labor laws, especially the legal policies of unemployment and healthcare benefits. First, it builds an ontology of job skills to match a *curriculum vitae* (CVs) and job descriptions (JDs). Second, an ontology for representing labor law documents is also proposed to support users in searching for legal regulations about labor law. The integrated ontology is used to construct the knowledge base of a search system for job seeking and labor law searching. This study also builds an intelligent resume-searching system in information technology (IT). This system is also combined with questions about Vietnamese labor laws on unemployment

and healthcare benefits. According to the experiment, the designed method effectively supports job seekers and users searching for legal labor documents.

The next section presents several related works on methods for building ontologies of job skills and legal documents and problems of searching for resumes and legal documents. 'Knowledge Base for Recruitment with Law Search System' describes the structures of ontologies for representing the content of resume applicants and JDs and labor law documents. The architecture of the IT recruitment system that combines the search for labor laws is also presented in this section. Next, 'Search System for IT Job Applicants and Labor Laws' discusses several problems of the searching system for resumes and labor law documents. Based on matching extracted skills from CVs and JDs, this technique is used to provide top k-CVs rankings for a specific JD. The designed system also has the ability to search for Vietnamese labor laws. 'Experimental Results' provides the results of the experiments for the resume searching system on the IT domain combined with searching for the Vietnamese labor laws. The conclusion section is presented in the last section and gives directions for future work.

## RELATED WORK

Many studies have been carried out to address the problem of screening CVs for a job description. The study suggested many automated systems with different techniques. *Najjar, Amro & Macedo (2021)* studied an automated approach for intelligent applicant screening for hiring by using ontology mapping. In order to suggest the best resumes for job postings utilizing a similarity assessment, their research concentrated on developing ontology papers for candidates' difficulties. By identifying broadly common concepts, concept linking was utilized to connect data and documents, where data models are ontologies. To categorize semantic linkages between related components of various ontologies, they used ontology mapping. They also developed an ontology of candidate qualities to match data from a CV and a JD. However, this study's capacity to extract abilities from this text is rather restricted. It does not precisely extract the information required to develop an ontology in the real world.

Another strategy is to create an electronic recruitment system with automated personality mining and application ratings. *Faliagka, Tsakalidis & Tzimas (2012)* offered a method for electronic hiring that automatically ranked candidates according to a set of reliable criteria, making it easier for businesses to integrate with their pre-existing human resources (HR) administration infrastructure. That technology was created to execute automatic candidate rankings based on objective characteristics acquired from the candidates' LinkedIn profiles. Individual selection criteria are used to generate the ranking of applicant profiles using the analytical hierarchy process (AHP), while the recruiter controls their relative relevance (weight). Although this study demonstrated a viable technique for mining applicant personality, it did not extract talent from the candidate profile information. As a result, it is quite difficult to apply for IT recruitment since IT candidates often possess certain required abilities for future work growth. Furthermore, transfer learning is a valuable machine learning approach for dealing with language problems (*Pillai & Sivathanu, 2020*; *Nguyen*

*et al., 2020a*) and particularly for extracting key terms from documents (*Loyarte-López & García-Olaizola, 2022*). To extract applicants' abilities, *Loyarte-López & García-Olaizola (2022)* used ontology and word embedding. These researchers offered a theory on matching candidates with organizational cultures to provide a better candidate who is a good fit for the company. Besides, *Yu, Palefsky-Smith & Bedi (2016)* used a policy gradient-based reinforcement learning technique to mimic the control of an autonomous automobile. However, such approaches could not be used to filter CVs automatically, and the findings were ineffective for developing a practical recruiting system.

The studies in *Baad (2019)* and *Kim, Kim & Kim (2019)* used recurrent neural network models to capture the context of skill phrases to build a taxonomy of occupational skills for recruitment systems automatically. These studies also implemented certain system approaches, such as long short-term memory (LSTM), bidirectional long short-term memory (BiLSTM), and conditional random field (CRF). Furthermore, a few rule-based approaches were employed to extract skills listed in front of preceding sentences, signifying the beginning of the skills from a JD. However, these skills were added as another tag by their Part-Of-Speech (POS) tagger; the technique missed many skills and was not able to detect skills from unstructured JDs. Moreover, the technique was ineffective for extracting abilities from CVs since the structure of a CV differs from that of a JD. A method to substitute specific hidden values inside datasets and inductive learning techniques was put forth by *Ravita & Rathi (2022)*. They did this by using a graph-based recommendation system, which helped systems' constraints, such as data sparsity issues, be handled using linkage exploration methods from the graph model. This item enhances the effectiveness of the recommendations. *Wang, Allouache & Joubert (2021)* constructed a knowledge graph (KG) from the structured competence map, which can classify the relationships of bidirectional association and aggregation. After that, it used named-entity recognition (NER) and masked language modeling (MLM) of BERT to better identify tokens from the input inquiries of the client. From that, it can find suitable candidates through analyzing the CV dataset. The knowledge graph composed of a competency map and competency keywords can recommend good results in the neighborhood domain.

There are many regulations related to labor law that affect workers. Therefore, it is crucial to process legal documents intelligently. Various methods have achieved significant advancements in intelligent legal document processing, including legal document extraction, categorization, and abstract document extraction (*Zhao, Liu & Erdun, 2022*; *Mirończuk, 2018*). LIDO is an ontology for a legal informatics document (*Sartor et al., 2011*). The structure of the legal resource, the legal temporal events, the legal activities impacting the document, and the semantic organization of the legal document can all be represented by this ontology. This is built based on the standard CEN Metalex. *Huynh, PhamNguyen & Do (2019)* developed a mechanism for combining a database of document repositories with an ontology that describes domain knowledge. This technique uses a domain knowledge model known as the classified key phrase-based ontology for a variety of information retrieval tasks (CK-ONTO). The practice of query reformulation is a recognized method aimed at enhancing the effectiveness of Information Retrieval Systems. Among the various techniques at hand, Query Expansion involves rephrasing the original

query by incorporating related terms sourced from various outlets such as corpora and knowledge bases (*Selmi, Kammoun & Amous, 2022*). This augmentation aims to enhance the retrieval of documents that are more pertinent to the query. In general, the semantic relevance of legal documents has not been assessed using the graph-based measure in this model. By combining the strengths of conventional information retrieval techniques, pretrained masked language models (BERT), and expertise in the legal domain, *Huy et al. (2021)* developed a system for Vietnamese legal text processing. A novel data augmentation technique based on legal domain expertise in legal textual entailment was also suggested. The suggested approach, however, does not capture the legal document's meaning.

*Li et al. (2020)* distinguished resumes by levels of competence. Additionally, it evaluated resumes and JDs to see if applicants were qualified for specific positions, which can considerably reduce the heavy daily labor load performed by HR recruiters. *Gugnani & Misra (2020)* combined many natural language processing (NLP) methods to suggest a method for skill extraction, and the concept of extracting implicit skills was then introduced in the study. Implicit skills are those that are not expressly addressed in a JD but may be implied in the context of geography, industry or function. Finding other JDs that are comparable to this JD allows us to mine and deduce implicit skills for a JD. This similarity check is carried out in the semantic domain. The summarization and transformer architectures were used to profile resumes (*Bondielli & Marcelloni, 2021*). This approach generated resume embeddings and used hierarchical clustering algorithms for grouping those embeddings. This study evaluated different strategies on a public domain dataset containing 1,202 resumes. The inter-résumé proximity method in *Cabrera-Diego et al. (2019)* was the lexical similarity between only resumes in response to the same job offer. Relevant feedback is also applied to similarity coefficients and vocabulary scoring to improve resume ranking. *Phan et al. (2021)* investigated how the JD/CV matching process can be improved by combining contemporary language models (based on transformers) with knowledge bases and ontologies. That approach sought to use knowledge bases and features to support the explicability of JD/CV matching. Finally, we propose a fair, explicable, and traceable architecture for JD/CV matching purposes, given the exploration of various software components, datasets, ontologies, and machine learning models. The chatbot in *Nguyen et al. (2020b)* was created to guide users with a variety of administrative processes, including acquiring a printing license. Nevertheless, this approach is unable to facilitate searching through a legal document for information relevant to the working domain.

Graph neural networks (GNNs) and statistical relational learning (SRL) are two potent methods for learning and inferring graphs. They are typically assessed using straightforward metrics, including accuracy on individual node labels. In order to compute the values of aggregate graph queries in a tractable manner, *Embar, Srinivasan & Getoor (2021)* developed a sampling methodology (AGQ). Such an approach is ineffective for arranging the meaning of a legal document because it only organizes data from social networks. The pursuit of recruitment opportunities and legal inquiries within labor law currently exists as discrete functionalities within distinct programs. An integration of both functions is presently absent in available programs. In practical scenarios, individuals engaged

in job-seeking activities may face unemployment or aspire to transition to alternative employment. Consequently, a necessity arises for users to investigate regulations concerning unemployment insurance policies and pertinent information and concurrently explore regulations regarding the smooth transition of employment, all while ensuring compliance with labor laws. Consequently, the paramount contribution of this study lies in establishing a foundational framework for designing ontologies capable of amalgamating both functions within a unified system. This system is envisioned to provide comprehensive support to workers engaged in job searching and labor law exploration.

# KNOWLEDGE BASE FOR RECRUITMENT WITH LAW SEARCH SYSTEM

## Ontology for organizing job applicants and job recruitments

Ontology is the essential paradigm for organizing information that fulfils the requirements of a recruiting system (*Chandrasekaran, Josephson & Benjamins, 1999*). In the semantic web model, the meaning of intended information, such as data or knowledge, is addressed using ontology (*Calaresu & Shiri, 2015*). Ontology also goes beyond the semantic web's most basic idea by granting capabilities to conventional reasoning, which frequently rests on the declaration of inference rules. In the resume-searching system, ontology is used to represent the data that was taken from applicants' CVs and recruiters' JDs.

**Definition 3.1**: The Job-Onto ontology structure represents job applicants and job recruitments as a triple:

$$(\mathbb{CV}, \mathbb{JD}, \mathbb{R}e),$$

where

- $\mathbb{CV}$ denotes the set of applicants' *curriculum vitae*,
- $\mathbb{JD}$ denotes the set of job descriptions, and
- $\mathbb{R}e$ denotes the set of relationships between job applications and job descriptions.

Standard ontology languages, such as OWL, can be used to build ontology documents.

## Structure of a *curriculum vitae*

Each *curriculum vitae* $c$ of an applicant, $c \in \mathbb{CV}$ has a structure as follows:

$$c = (Profile, History\_work, Edu, Courses, Skills, Position, Others),$$

where

- *Profile*: stores the applicant's personal information.
- *History_Work*: These were the applicant's positions at earlier firms.
- *Edu*: applicant's educational background.
- *Courses*: It comprises certain additional training courses completed by the candidate. These courses provide more information regarding the candidate's comprehension and self-study.
- *Skills*: the information about the candidate's skills. There are two kinds of skills:

- *Domain skills*: they store the major skills of the applicant in IT. They are divided into two categories: programming environment skills and general programming abilities.
- *Soft skills*: they are some of the applicant's soft skills, such as teamwork, communication skills, leadership, and logical and critical thinking.

- *Position*: the applicant's application position.
- *Others*: other applicant information.

## Structure of a job description

Each job description $d \in \mathbb{JD}$ has the following structure:

$$d = (Infor, Descript, Req, Skills)$$

where

- *Infor*: general information about the recruited position.
- *Descript* (Descript $= \{s_1, s_2, \ldots, s_m\}$): set of sentences that describe working at the recruited position. Each sentence $s_j$ $(1 \geq j \geq m)$ is a set [*text, keywords* ], where text is a sentence of text to describe the job, and keywords are a set of keywords extracted from text.
- *Req*: detailed requirements for a candidate.
- *Skills*: the skills for the recruited position. The categories of talents are organized similarly to a resume. The system may automatically enter this component from the recruiter or derive it from data in the Descript and Req components.

## Ontology for representing legal knowledge domain about labor

When users seek suitable jobs, they search for several legal documents regarding labor and work to understand more about labor regulations. This section presents an ontology representing the law documents about labor and work to search for this legal domain. The proposed ontology is an integrated structure of the knowledge model of relations, Rela-model (*Do, Nguyen & Selamat, 2018*; *Nguyen et al., 2021*), and the conceptual graph representing the linking of key phrases in law documents (*Filtz, 2017*).

Vietnamese labor law documents include the Law of the National Assembly and sub-law documents. The Law of the National Assembly is a general rule of conduct, commonly binding and repeatedly applied to agencies, organizations, and individuals nationwide in the labor field. Sublaw documents are legal files that describe the details of the law established by the National Assembly. The built legal ontology has to represent the meaning of those documents and the links between their articles.

**Definition 3.2:** Ontology representing a legal document, called Legal-Onto, is built based on the Rela-model. It includes the following components:

$$\mathbb{K} = (\mathbb{C}, \mathbb{R}, \mathbb{RULES}),$$

where

- $\mathbb{C}$ is the set of concepts, but each concept in $\mathbb{C}$ has improved its internal structure to organize its law information;

**Table 1  Structure of the concepts.**

| Name & Content | Rel-Inner | Attrs | Phrases |
|---|---|---|---|
| National Certificate of Vocational Skills: A national certificate of vocational skills is a certificate that means a worker can perform satisfactory jobs at a skill level of a profession. | *Decree on Detailing Unemployment Insurance 2015*: Article 33 | Object, Condition | Vocational skills |
| Unemployment insurance: Unemployment insurance is a system for compensating a portion of an employee's salary when he or she loses a job, as well as assisting workers in vocational training, job maintenance, and job seeking, all of which are supported by contributions to the Unemployment Insurance Fund. | *Labor Code 2019*: Article 3, Article 43, Article 45, Article 57 | Time, benefit, insured person, unemployment insurance fund | Unemployment insurance, salary, benefit, lost job, vocational training, job maintenance, job seeking |

- $\mathbb{R}$ is the set of relations, and those relations are between concepts, key phrases, and the database storing the legal document's content and
- $\mathbb{RULES}$ is the set of inference rules on the domain of labor laws.

## $\mathbb{C}$—the set of concepts

The law contains general principles based on widely accepted concepts. In the real world, in addition to the content, the structure of each concept in a legal document has to present relationships between itself and the articles in the corresponding document.

**Definition 3.3:** Given a law document d as Legal-Onto, each concept $c \in \mathbb{C}$ consists of five elements:

$$c = (Name, Meaning, Rel\text{-}Inner, Attrs, Phrases)$$

where

- *Name*: The legal concept's name.
- *Meaning*: The meaning or the content.
- *Rel-Inner*: a list of the document's articles connected to the relevant concept.
- *Attrs*: List of elements (or other concepts) that support the corresponding concept in document d, if one is necessary.
- *Phrases*: the list of keywords describing the meaning in each document article $d$.

**Example 3.1:** The structure of the concepts in the National Certificate of Vocational Skills (*Government of Vietnam, 2015*) and unemployment insurance in *Law on Employment 2013* are organized as in Table 1.

## $\mathbb{R}$—the set of relations

The set $\mathbb{R}$ is a set of relations. Each relation in $\mathbb{R}$ is one of three kinds:

$$R = R_{con} \cup R_{key} \cup R_{data}$$

where,

**Table 2  Four types of facts.**

|   | Meaning | Specification |
|---|---------|---------------|
| 1 | Display a property of a relation | $[< relation > is < property >]$ $relation \in \mathbb{R}$ |
| 2 | Relations between concepts. | $[< c_1 >< relation >< c_2 >]$ $c_1, c_2 \in \mathbb{C}$ |
| 3 | Relations between key phrases | $[< k_1 >< relation >< k_2 >]$ $k_1, k_2 \in Key$ |
| 4 | Relations between key phrases and a concept | $[< k >< relation >< c.Phrases >]$ $k \in Key, c \in \mathbb{C}$ |

- $R_{con}$ is a set of relations between concepts in $\mathbb{C}$. These relations are "is-a" (hierarchical relation), "has-a", "a-part-of", and several other relations between concepts.

$R_{con} = \{r | r \subseteq \mathbb{C} \times \mathbb{C}\}$.

Several properties of each relation $r \in R_{con}$ are considered symmetric and transitive.

- $R_{key}$ is a set of connections between important terms in the legal document. Many relationships between a concept and key phrases, which are distinctive words that help define a concept's meaning, are also included in this set.

$R_{key} = \{r | r \subseteq Key \times Key\} \cup \{r' | r' \subseteq Key \times \mathbb{C}.Phrases\}$

- $R_{data}$ is a set of relations between concepts and key phrases linked to a database of the law document.

## $\mathbb{RULES}$—the set of rules

The regulations within the $\mathbb{RULES}$ embody limitations and the derivation of connections among fundamental words and concepts. Employing deductive rules during the construction of ontology data can streamline the responsibilities of a knowledge engineer. The $\mathbb{RULES}$-set infers both direct and indirect links between keywords or concepts, subsequently employed to establish semantic similarity (*Nguyen et al., 2022b*). A rule $rul \in \mathbb{RULES}$ conforms to a deductive pattern grounded in information related to pivotal words and concepts, characterized as follows (*Nguyen et al., 2022b*):

$$rul : \{h_1, h_2, \ldots, h_m\} \rightarrow \{t_1, t_2, \ldots, t_n\}$$

where $\{h_1, h_2, \ldots, h_m\}$ are hypothesis facts and $\{t_1, t_2, \ldots, t_n\}$ are goal facts of rule $rul$. There are four kinds of facts as shown in Table 2.

**Example 3.2:** Several rules in the knowledge domain:

$r_1$: if [ $\Omega$ is symmetric] and [ $k_1 \Theta k_2$], then [ $k_2 \Theta k_1$] ($\Omega$ is a relation, $\Omega \in \mathbb{R}$)

$r_2$: if [ $\theta$ is transitive] and [ $k_1 \theta k_2$] and [ $k_2 \theta k_3$], then [ $k_1 \theta k_3$] ($\theta$ is a relation, $\theta \in \mathbb{R}$)

## Keyphrase graphs of law documents

Knowledge of a law document is a keyphrase graph, which is organized using the tube $(\mathbb{C}, \mathbb{R}, \mathbb{RULES})$ as the structure of the Rela model. While getting legal content, several key phrases in the query have been linked to the information through their semantics. An improved conceptual graph is used in this study to structure the semantics of key phrases.

The structure of the Rela model organizes the knowledge of a law document. However, in practice, when retrieving the content of the law, several key phrases in the query sentence

have been connected to the knowledge through their semantics. In this study, the semantics of key phrases are organized by an improved conceptual graph.

**Definition 3.4:** The structure of the key phrases graph of law document $d$ is a tube:

$$\mathbb{KG} = (Key, Rel, w_k, w_e)$$

where

- $Key = \{k|k$ is a key phrase of document $d\}$.
- $Rel = \{e = (k_1, k_2) \in Key \times Key|k_1$ and $k_2$ are key phrases that appear in the same law document article$\}$.
- $w_k: Key \to \mathbf{R} \times \mathbf{R}$ is the weighted map to compute the similarity binary vector for each key phrase in Key ($\mathbb{R}$ is the set of real numbers). The vector $(tf(p,d), idf(p,d))$ computes the key phrase similarity measure, where $tf(p,d)$ is the term expressing the frequency of a keyphrase $p$ in a document $d$, and $idf(p,d)$ is the inverse article frequency showing the specificity of the document $d$'s keyphrase p. The following is how the formulas for $(tf(p,d), idf(p,d))$ are calculated (*Huy et al., 2021*):

$$tf(p,d) = c + (1-c)\frac{n_{p,d}}{\max\{n_{p',d}|p' \in Key\}} \tag{1}$$

where $n_{p,d}$ is the number of occurrences of the key phrase p in document $d$, $c \in [0,1]$ is a parameter that represents the minimum value for each key phrase.

$$idf(p,d) = \log\left[\frac{card\left[\bigcup_{ar \in d} key(ar)\right]}{1 + card(A_p)}\right] \tag{2}$$

where, $A_p := \{ar \in Arctics(d)|p \in key(ar)\}$ Artics(d) is the set of articles of the law document d, key(ar) is the set of key phrases of article ar in document d.

- $w_e: Key \times Key \to \mathbf{R}$ is the weighted map to compute the similarity binary vector for each key phrase in Key (**R** is the set of real numbers).

$$w_e(p,p') = \frac{card(B_{p,p'})}{card(Arctics(d))} \tag{3}$$

where,

$$B_{p,p'} := \{ar \in Arctics(d)|p,p' \in key(ar), r_{ar}(p,p')\}$$

key(ar) is the set of key phrases in article ar, $r_{ar}(p,p')$ is the relation between $p$ and $p'$ in article ar.

## Ontology and keyphrase graph integration

Legal-Onto ontology is used to represent the information within a legal document. The utilization of a knowledge graph is efficient in establishing connections among various components in a legal document. Consequently, achieving a comprehensive model for depicting the content of a legal document needs the combination of both Legal-Onto and the knowledge graph. The combination model is described as follows:

$$\mathbb{K} = (\mathbb{C}, \mathbb{R}, \mathbb{RULES}) \bigoplus (Key, Rel, w_k, w_e).$$

The symbol $\bigoplus$ means the ontology $(\mathbb{C}, \mathbb{R}, \mathbb{RULES})$ as Rela-model is integrated with the conceptual graph $(Key, Rel, w_k, w_e)$. The integration is worked through relations between the key terms of concepts in $\mathbb{C}$ and key phrases in Key.

The key phrase graph and ontology $(\mathbb{C}, \mathbb{R}, \mathbb{RULES})$ as the Rela-model are integrated through relations between key phrases in Key and key terms in $\mathbb{C}.Phrases$. These relations are represented in $R_{Key}$.

**Definition 3.5:** Given an ontology $(\mathbb{C}, \mathbb{R}, \mathbb{RULES})$ as a Rela-model and a key phrase graph $(Key, Rel, w_k, w_e)$. Integrating them works based on relations in the following set:

$$Integ\_rel = \{r' \in \mathbb{R} | r' \subseteq Key \times \mathbb{C}.Phrases\}. \tag{4}$$

Thus, $Integ\_rel \subset R_{Key}$

From an input query, the key phrase graph helps to quickly extract the semantics in legal documents. Matching the meaning of that query with the knowledge of legal documents works based on relations in $Integ\_rel$. Through that, the process is able to retrieve suitable contents from corresponding law documents.

The knowledge base, represented by the integration of ontology Legal-Onto and knowledge graph, can be applied to build an intelligent recruitment system with the law search system. The section presents the utilization of designing a search system for job applicants in information technology (IT) and Vietnamese labor laws.

# SEARCH SYSTEM FOR IT JOB APPLICANTS AND LABOR LAWS

This section presents the construction of an intelligent searching system for IT job applicants, combining queries on Vietnamese labor laws. This system has two main functions: (1) searching candidates (or jobs) suitable for an input JD (or CV) in the IT domain, and (2) querying Vietnamese labor law documents, especially the policies of unemployment and healthcare benefits.

## Architecture of IT recruitment system

The CSO classifier is an unsupervised technique used by the IT recruiting system to categorize documents based on ontology-based content (*Salatino et al., 2019*). The syntactic module and the semantic module make up this technique. The Job-Onto ontology represents the information content of CVs and JDs *via* the CSO classifier. In addition, this classifier also improves the processing of the labor law document to extract key phrases from this document. They are material to build the Legal-Onto ontology for representing labor laws. The workflow of the IT recruitment system combining the searches for labor laws is shown in Fig. 1.

The functions of several modules in the CSO classifier are as follows:

- Syntactic module: This module associates textual n-gram chunks with concepts. The input text is eliminated the stop words before extracting chunks of unigrams, bigrams, and trigrams are collected. The Levenshtein distance similarity (*Arockiya Jerson & Preethi, 2023*; *Government of Vietnam, 2015*) with the topic labels in the ontologies is

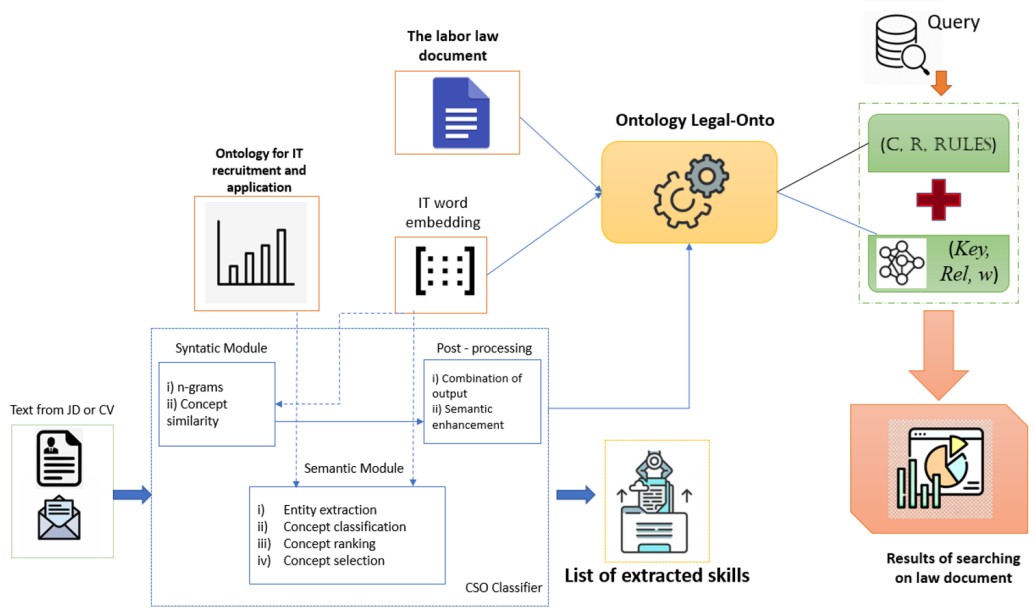

**Figure 1** The architecture of the IT recruitment system combining the search for labor law.

then calculated for each n-gram. The minimum similarity level has been manually set at 0.94. We can distinguish a wide range of distinctions between ideas and ontologies.

- Semantic module: The semantic module was created to identify subjects in the text that are semantically connected. The text makes no mention of such matters. To determine how semantically similar the terms in the text are to those in the ontologies, they use Word2Vec's word embeddings.

This study uses the Word2Vec model (*Salatino & Osborne, 2019*; *Gao et al., 2022*) to construct word embeddings for preprocessing collected data. First, the model is trained for text collection from the CVs and JDs in the IT domain. The Job Posting dataset is collected from Dice.com (https://www.dice.com/) and StackOverflow (https://stackoverflow.com/). ESCO (https://ec.europa.eu/esco/portal) is used for an alternative name and skill description. Finally, 50 thousand sentences containing extracted skills are preprocessed from the Job Post in the Skill2Vec (https://github.com/duyet/skill2vec-dataset) dataset. After the training process, the classifier can extract skills for an input JD or CV. Second, in addition to IT skills, several key phrases in the IT domain and labor have been deduced from the training results. From the labor law document, several key phrases in the labor law are also extracted. All of the produced key phrases are material for building the ontology Legal-Onto to represent the corresponding law document. They are also used to process the query on the meaning of the law document represented by ontology Legal-Onto.

## Creating ontologies
The CSO classifier's documentation is used to create the domain-specific IT ontology from the job postings manually.

- Building ontology Job-Onto in the IT domain: There are 10,000 job listings for each branch of IT, such as frontend developer. Each job post has all of the keywords. After removing all stop words and unrelated terms, a set of skills extracted from all keywords is built by mixing them with Stackshare skills. The domain-specific ontology is constructed using the Protégé program once all of the abilities had been explored. The domain-specific IT ontology was manually generated from the job listings based on its documentation to input them to the CSO classifier.
- Building ontology Legal-Onto for Labor Laws: The legal ontology about labor is created from the collected labor law documents. This ontology includes knowledge components about concepts, relations, and inference rules for this domain. In addition, several common questions and answers for querying labor knowledge are collected from *Nguyen (2021)*. Based on those, the list of key phrases in labor law is extracted and represented as a conceptual graph connecting to the articles of the documents. The created ontology is used to organize the semantics of the corresponding document and tends to search for the labor knowledge of legal documents.

Figure 2 displays the process of creating ontologies for this search system:
- **Step 1:** Define resources of the knowledge domain; then collect data/knowledge from them.
- **Step 2:** Extract key phrases from crawled data
- **Step 3:** Build the content of ontology

  - **Step 3.1:** Determining the structure and knowledge model. The data conforms to the RDF schema serving the Rela-model.
  - **Step 3.2:** Convert the collected data according to the specified structure. Construct and initialize classes; the classes are extracted keywords.
  - **Step 3.3:** Forming the attributes of each class. Each class will have its own characteristics to distinguish classes and represent the semantics of that concept more effectively. Create hierarchical or peer-to-peer relationships between classes by creating hierarchical or peer-to-peer interactions.

## Resume search system in information technology
### Architecture of the searching system
The resume search system can rank CVs for an input JD and recommend JDs for an input CV. This method implements automatic CV ranking using a set of criteria related to general technical skills. Those skills are extracted from a CV, domain technical skills, domain skills that the JD is looking for, and soft skills. The operation of the resume searching system is described in Fig. 3.
There are several modules in this work:

- Resume parser module: This module takes the uploaded CV and parses it into text for skill extraction.
- Extracting skills module: This module extracts a list of abilities from the JD and CV by categorizing them into four categories: general technical skills, education, domain skills, and soft skills. CSO-Classifier, an unsupervised technique for autonomously categorizing

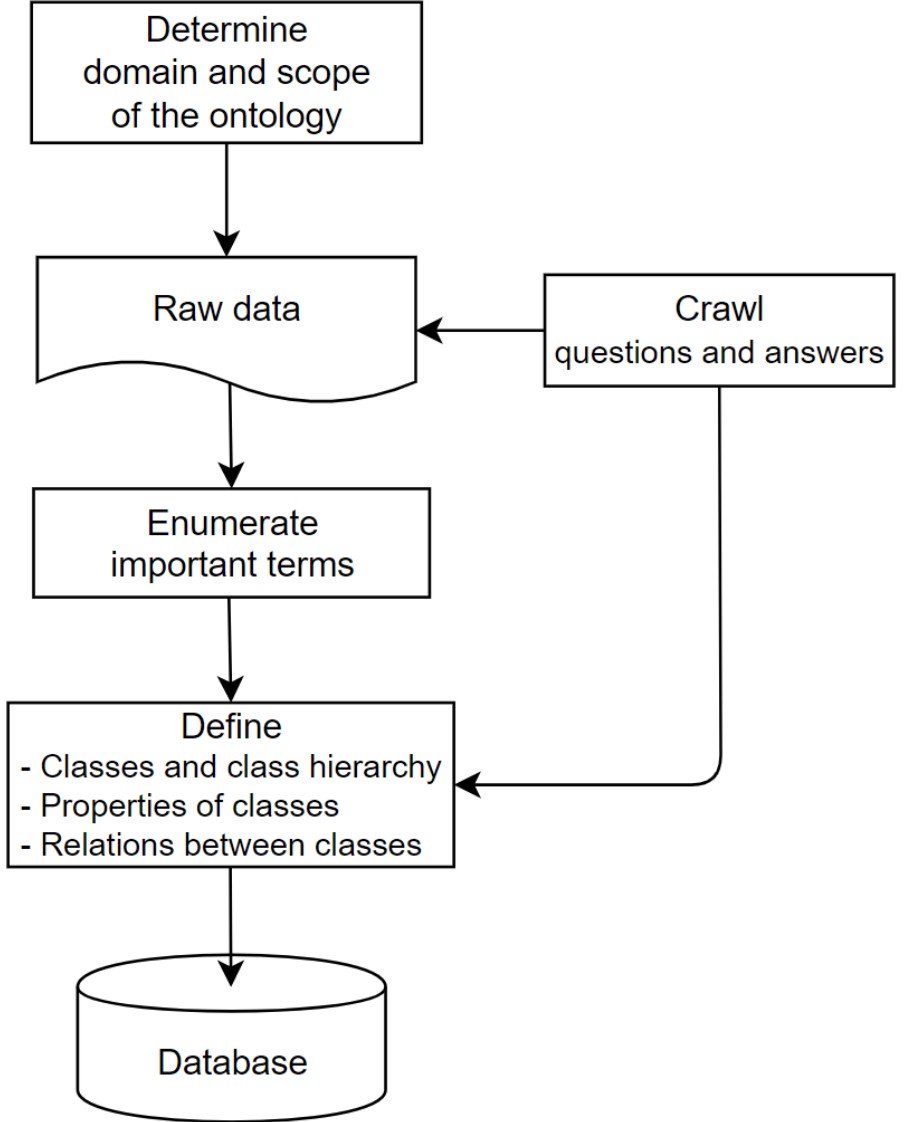

**Figure 2** The process for creating ontologies.

papers according to the Job-Onto filter (*Salatino et al., 2019*), is used. The extracted skills are represented by a skill graph.

• Matching module: This module uses candidate or JD selection criteria to calculate the relevance score of each criterion for the applied post. The grading function is built using the extracted skills module's determined similarity of skill graphs. The matching process is presented in the next section.

Moreover, when searching for suitable resumes, the system permits the recruiter to add the weights assigned to the various needed abilities, and the system displays a list of candidate resumes that have been ranked in accordance with their overall ratings.

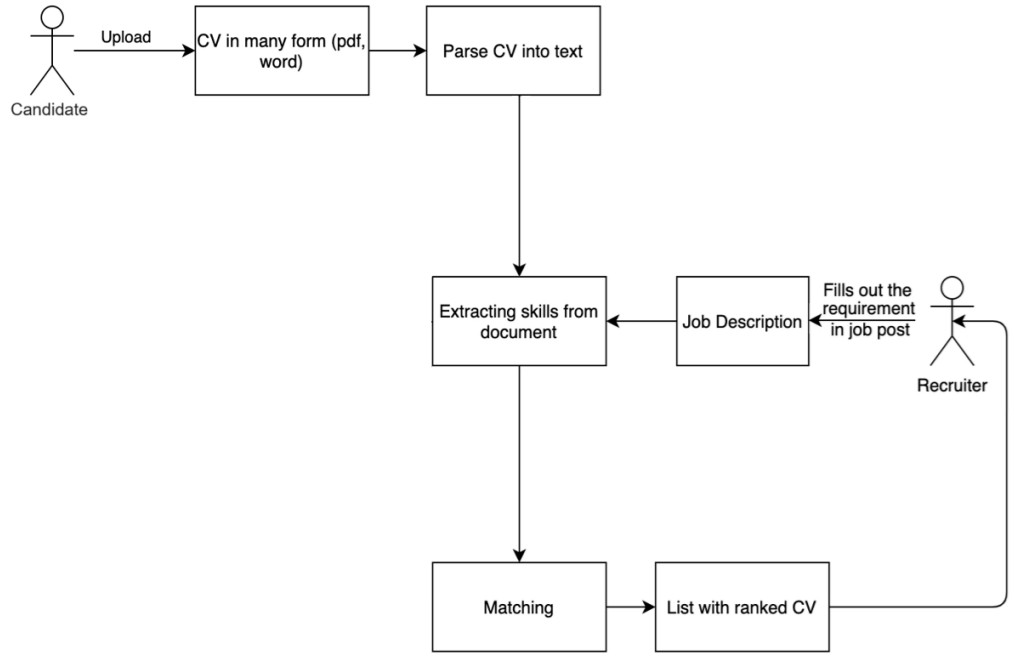

**Figure 3** **The matching process between CVs and JDs.**

## The method for matching CV and JD

Data collection: There are several datasets available on the internet, but selecting one that fits our needs is exceedingly tough. First, we gather a substantial number of job postings from a variety of employment websites, including Dice.com, Indeed, and others. We extracted the job description from each job posting and cleaned the raw data before feeding it into our word2vec model, detailed later in this section.

We gather and prepare a large dictionary of skills, as recommended in *Gugnani, Kasireddy & Ponnalagu (2018)*, to perform decent skill extraction. This talent dictionary is created by scouring many websites for information on skill terms and field terminology (ESCO, StackShare; https://www.stackshare.io). We are able to create a dictionary containing over 30,000 skill phrases after cleaning and analyzing these terms.

Data processing: extracting text from a CV. First, we assume that parsing a CV from a PDF or Word document into raw text would be straightforward; however, we are mistaken. Because a CV is an unstructured document, building a parser for it is extremely difficult. We test PyPDF2, PDFMiner, and PDFtoTree, as well as other existing tools for parsing PDF formats into text. One frequent blunder made by all the preceding programs is that the extracted text is occasionally placed in the wrong location. Optical character recognition (OCR) is therefore used in the suggested technique. We attempt to parse a CV into text using this method, which includes the steps below:

- Creating pictures from the CV; each image represents a single page of the CV.
- Using OpenCV (https://opencv.org/), create a threshold image.

- Using tesseract (https://github.com/tesseract-ocr/tesseract) OCR to parse text from a threshold picture.

This method has various drawbacks; for example, it takes longer to extract text straight from the source file, and the quality of the retrieved data is poor if the scanned pages are blurry. However, after testing approximately 1000 CVs, this technique yields positive findings that can be used in practice.

### Pre-processing data for the Word2Vec model

The gathered data has a variety of structures due to Because the data is gathered from many sources. Therefore, the pre-processing will be somewhat altered. The following stages will often be used to process data:

- Eliminate any HTML tags you find.
- Rearrange sentences into paragraphs.
- Eliminate special characters, but keep a few common in programming languages, such as "+", "#", and "C#".
- Eliminate stop words, which frequently exist in English literature but don't provide skill phrases any meaning (*e.g.*, a, the, and, or, it, if).
- Lemmatize each sentence.

After separating sentences and word processing, we have a dataset of 1.8 million sentences suitable for training the Word2Vec model.

### Extracting skills from CV and JD

The CSO classifier must extract the top 10 related terms for all tokens and compare them to CSO subjects while studying a text, which is a laborious process. A cached model for each domain is used to solve this problem. All tokens in the model's vocabulary are directly linked to skill terms in the ontology by the cached model, which serves as a dictionary. This item makes it easier to rapidly find all skill phrases implied by a certain token.

Next, a dictionary is created by placing the cached models in it. This extraction module's user must first select a domain before being able to extract skill phrases from the text, after which a CSO classifier for that domain is generated and utilized for extraction. The technique of obtaining skill terms is briefly depicted in Fig. 4.

### Skill graph generation

For feature matching, the extracted skill list is used to generate the skill graph. The domain ontology determines the edges between nodes drawn using the NetworkX graph library. The tree-like created network contains a root that is the domain ontology's name and relevant skills nodes. The following figures show an example of skill graphs generated from a CV (Fig. 5) and a JD (Fig. 6).

### Compatibility between CV and JD information

The graph edit distance (GED) is a similarity measurement between two graphs that we use to match features. In particular, the graph created by a CV will be updated to match the graph from JD using insert/update/delete activities. The graphs become more distinct

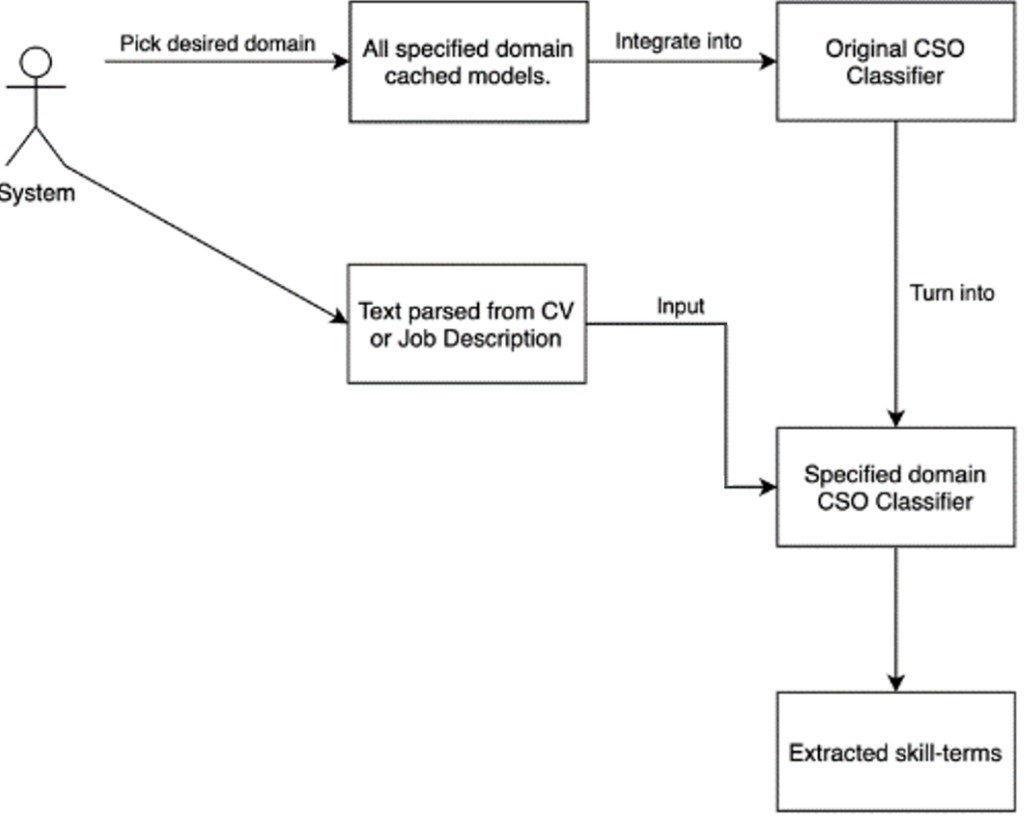

**Figure 4  Extracting skill terms for a specified domain process.**

as the number of transformation activities increases. Gmatch4py, a package specialized for graph matching, is utilized in this stage. It is calculated using a graph edit distance approach that combines Hausdorff matching and the greedy assignment (*Fischer, Riesen & Bunke, 2017*). The graph structures are saved as objects in the NetworkX graph.

The score of each criterion between CV and JD is obtained after the matching process. There are numerous sorting techniques to rank a set of CVs for each JD. Depending on the scenario and the demands of their firm now, recruiters may have their ideas about the characteristics they wish to emphasize. As a result, our resume search engine for job seekers provides such sorting by allowing recruiters to change the parameters as needed.

Figure 7 presents the recommendation processing of CVs and JD. When a candidate inputs his or her CV, the system parses this CV to text and extracts skills from that document. After that, the system creates a skill graph for the candidate and matches it with stored JDs in the system. In contrast, when a recruiter posts a JD, the system also extracts the required skills from the JD and creates a skill graph for the JD. Then, it matches this with stored CVs in the system.

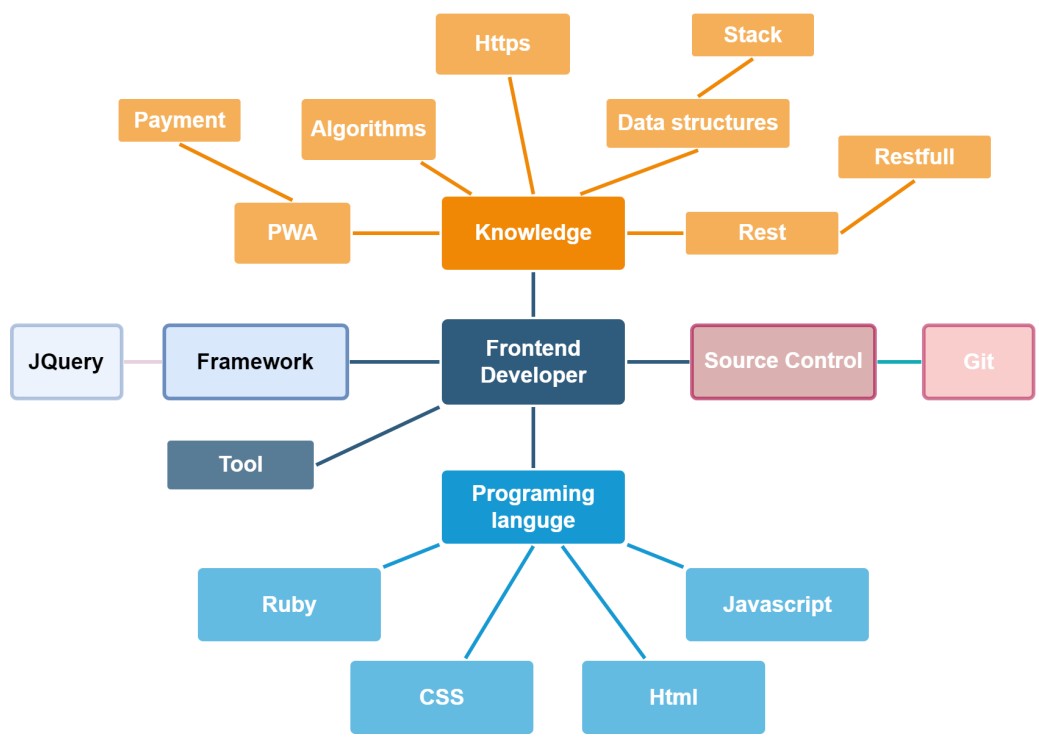

**Figure 5** The skill graph extracted from a CV.

## Search system on labor laws

Let $\mathbb{K}$ represent a knowledge domain of the ontology Legal-Onto model and the legal document $d$. The search system processes a search query to extract key phrases when one is entered to get knowledge from $\mathbb{K}$. These key phrases are matched with the appropriate content in the knowledge base based on their semantics. The matching is carried out by examining the relationships between key phrases that lead to corresponding concepts and the structure of the knowledge model's constituent parts. The knowledge base's inference rules aid in the matching process by enabling the deduction of additional relations pertinent to the query. The retrieval outcomes for the entered query are then shown. Several of the most common issues with finding legal documents are as follows: (1) Problem 4.1: The first issue is classifying the query that was entered. This problem takes the important keys of the query from the query supplied as Vietnamese text. Those phrases are used to determine the meaning of the query and classify it. (2) Problem 4.2: The second issue is finding relevant articles in the document and looking for concepts in the text based on keyphrase matches. The set of keywords obtained from the query is used to give a method for assessing how closely the meaning of the keywords matches that of the data in the knowledge base. This technique determines the knowledge material needed for the entered question.

Problem 4.1 has been solved in other studies, such as in *Do, Nguyen & Selamat (2018)* and *Nguyen et al. (2022a)*. This section only presents the solution of Problem 4.2. Following the extraction of the utterances of keywords and intents, the system will match those words and intents to the knowledge content to define and compare texts using the Legal-Onto

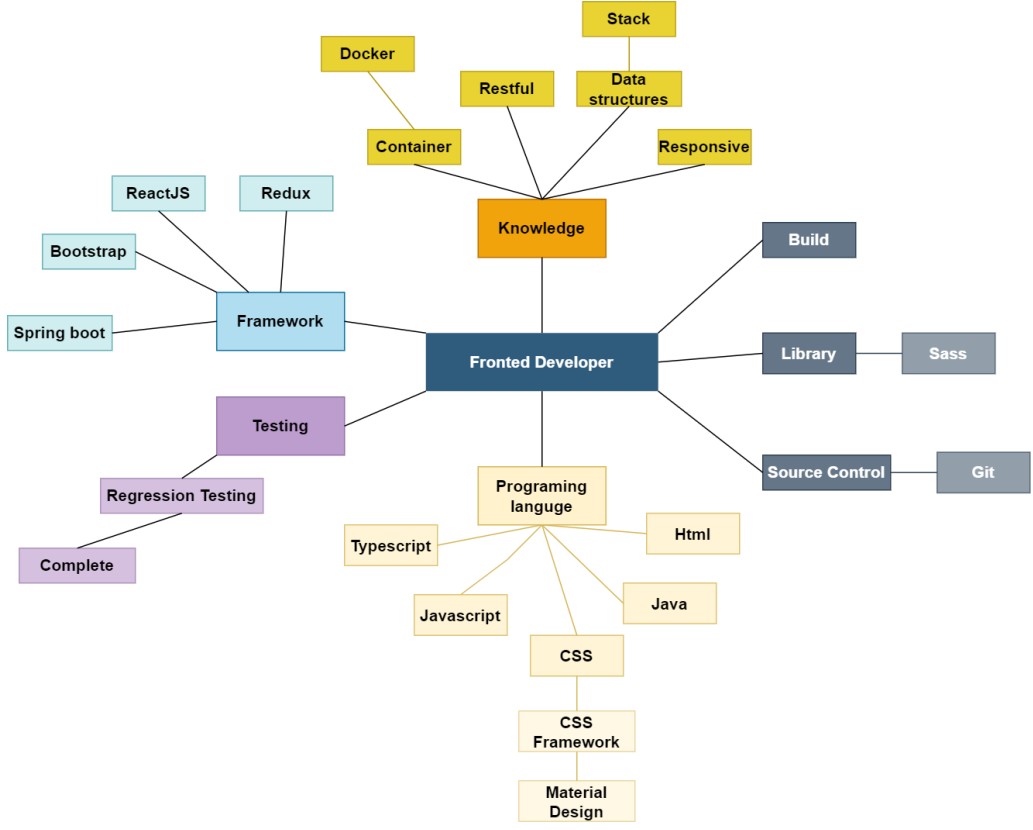

**Figure 6** **The skill graph extracted from a JD.**

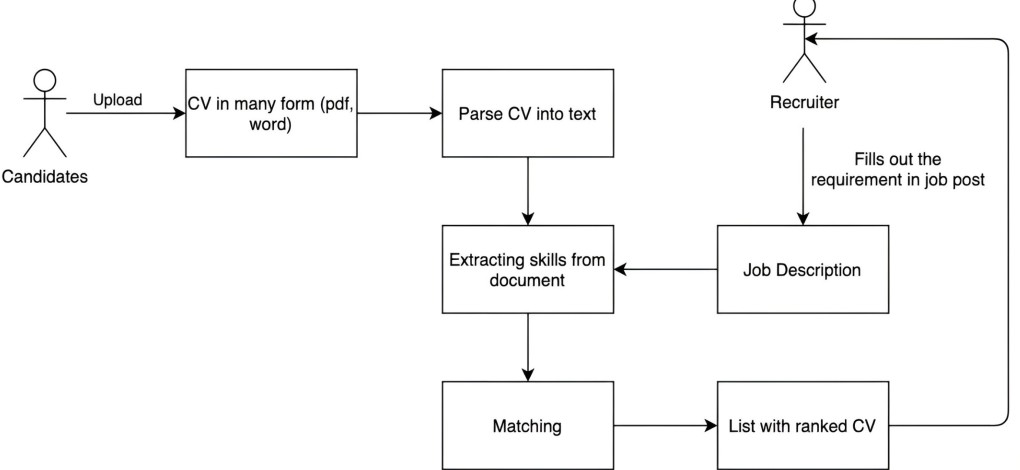

**Figure 7** **The recommendation process of CVs and JDs.**

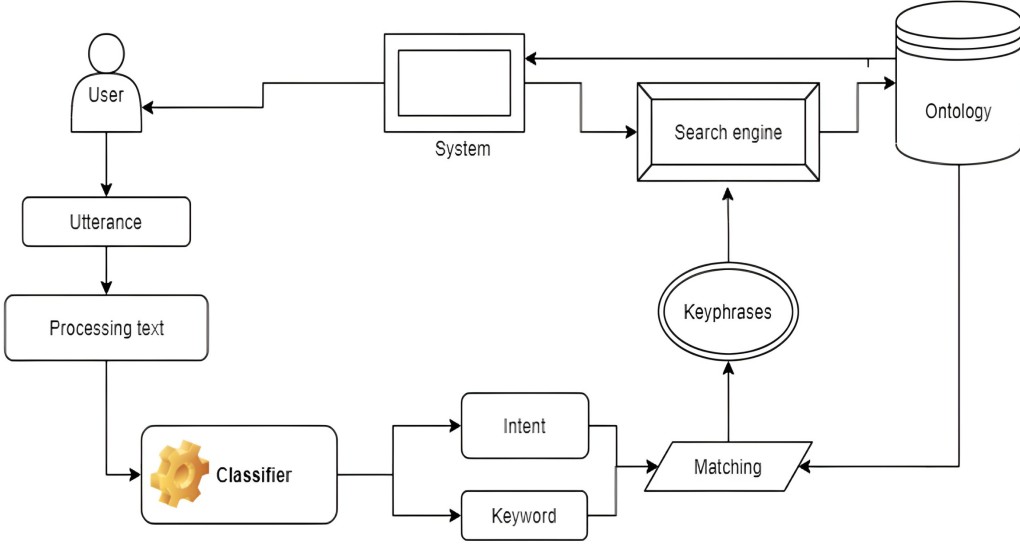

**Figure 8** **The matching technique for the search engine on labor laws.**

ontology as defined in Definition 3.2. Based on the approach in *Do, Nguyen & Selamat (2018)* and *Pham et al. (2020)*, the matching method for the search engine can be created, but it makes several improvements, as shown in Fig. 8.

The technique also extracts the query's primary key phrases after categorizing the input query. In Problem 4.2, these key phrases are used to find the appropriate information in the knowledge base. The keyword dictionary is created using the search engine's knowledge base and input from professionals in the field of law, including attorneys, senior legal staff, and senior law lecturers.

**Example 4.1:** Several key phrases in the dictionary of the Vietnamese Labor Code 2019:
- Several individuals in the dictionary: *representative organization, labor relation, forced labor, labor discrimination, sexual harassment, employment contract, indefinite-term employment contract, fixed-term employment contract, job- or position-based salary, form of salary payment, due date for payment of salary, allowances and other additional payments, regimes for promotion, regimes for pay rise, working hours, rest periods, protective equipment, social insurance, health insurance, unemployment insurance, basic training, advanced training, occupational skill.*
- The synonyms of the key phrases in the dictionary: "*what is*" is equivalent to "*define*", and "*how to use*" is equivalent to "*usage*".

The extracted keywords from the speech are compared with the dictionary to provide a collection of keywords. Keywords are used by the search engine to extract legal knowledge from stored knowledge. The system also proposes several relevant pieces of information from the search results based on the relationships between the acquired results and the knowledge base's inference rules. Finally, for each type of intent, all results are returned in the form of a document.

**Algorithm 4.1:** Finding matched knowledge of query $q$

**Input:** A query $q$.
**Output:** A set of knowledge content in document $d$
that matches the meaning of query $q$.
**Step 1:**
    **Extract** key phrases from the query sentence $q$
        W := key phrases(q)
    **Expand** $W$ based on relations in $R_{key}$.
    **Mapping** k in W to graph $\mathbb{G}$ as the key phrase graph
        (Key, Rel, $w_k$, $w_e$).
**Step 2:**
    *Known* := {} // set of results.
    *Concs* := {}; // set of concepts related to query $q$.
    **For** each *phrase* $\in \mathbb{G}$ **do**
        **Use** relations in $\mathbb{R}_{key}$ and rules in $\mathbb{RULES}$ for
        connecting *phrase* with concept $c \in \mathbb{C}$.
        **Update** $c$ into *Concs*.
    **For** each concept $c$ in *Concs*:
        **Update** $c$ into *Concs* based on relations in $\mathbb{R}_{key}$
        **Retrieve** knowledge from components of $c \in$ *Concs*
            **Update** *Known*.
**Step 3:**
    **Unification** of facts and compare their meaning.
    **Update** *Known*.
**Step 4:**
    **Return** results in *Known*

In Algorithm 4.1, a collection of knowledge is returned by the knowledge content search based on the meaning of an inputted query. The machine derives the meaning of this question from the terms it has retrieved. This search is carried out by finding query terms with comparable meanings and utilizing stored information.

## EXPERIMENTAL RESULTS

### Experiment on the resume search system

The CV and JD sets are gathered from Indeed and TopCV. The ontology Job-Onto serves as their representation. ReactJS and Flask are used to implement the resume-search system for IT job hopefuls. The number of collected CVs and JDs in our system is shown in Table 3.

Based on matching scores for domain skills, general skills, and soft skills for each posted JD, the algorithm calculates a correlation score for each candidate's CV. For instance, Fig. 9 and Table 4 display a list of ranked CVs for the JD of front-end engineers, along with overall scores. Table 2 is the score of the ten highest-score candidates matching the JD about the frontend developer role in Fig. 5B. These scores are computed based on the

**Table 3  The number of CVs and JDs entered into the system.**

| Order | IT fields | No. of CVs | No. of JDs |
|---|---|---|---|
| 1 | iOS Developer | 30 | 25 |
| 2 | Android Developer | 30 | 25 |
| 3 | Frontend Developer | 30 | 25 |
| 4 | Backend Developer | 30 | 25 |
| 5 | Fullstack Developer | 30 | 25 |
| 6 | Devops Engineer | 30 | 25 |
| 7 | AI Engineer | 30 | 25 |
| 8 | Data Science | 30 | 25 |
| | **Total** | **240** | **200** |

scores of domain skills, general skills, and soft skills. Those skills have weights from 1 to 3 (less important, important, and very important): "Domain" is 3, "General" is 3, and "Soft" is 1.

In practice, many methods can be used for matching CVs and JDs (*Tejaswini et al., 2022*; *Kostis et al., 2022*). However, Moreover, the three baseline methods, including Jaro–Winkler distance (*Boudjedar et al., 2021*), cosine similarity (*Kamacı, 2022*), and Levenshtein distance (*Arockiya Jerson & Preethi, 2023*), are usually applied in practical systems because their effectiveness and profit. Cosine similarity is a metric used to measure how similar two vectors are. Levenshtein distance is a straightforward method for measuring string similarity. When used to match a CV to a JD, these measures can be employed to assess the similarity between the skills and qualifications mentioned in the CV and those specified in the JD. Jaro–Winkler distance is a string similarity metric that measures the similarity between two strings. It is particularly effective for comparing short strings and can be used to compare individual skills extracted from JDs and CVs.

In this experiment, each inputted JD can be extracted for the content of general skills, domain skills, and soft skills. After that, those extracted kills are computed matching scores with CVs by cosine similarity, original Levenshtein distance, and the proposed method. Given a JD, there are two stages for comparing the results of the CV filter of each method for that JD:

- At first, establish lists of emerging CVs that were produced by each method.
- After that, there are two experts in IT recruitment building their lists of CVs for that JD. They will be asked to choose the Top 8, Top 5, and Top 3 of emerging CVs in their list. If there are differences between the two lists, another expert, who is the head of the Human Resources (HR) Department, will be asked to give the final decision.

Finally, the results of each method will be compared with the lists established by the experts. Table 5 compares the matching proportions of each method with the list built by experts in the Top 8, Top 5, and Top 3 for JDs.

The results of matching proportions of each experimented method with the list built by experts in cases.

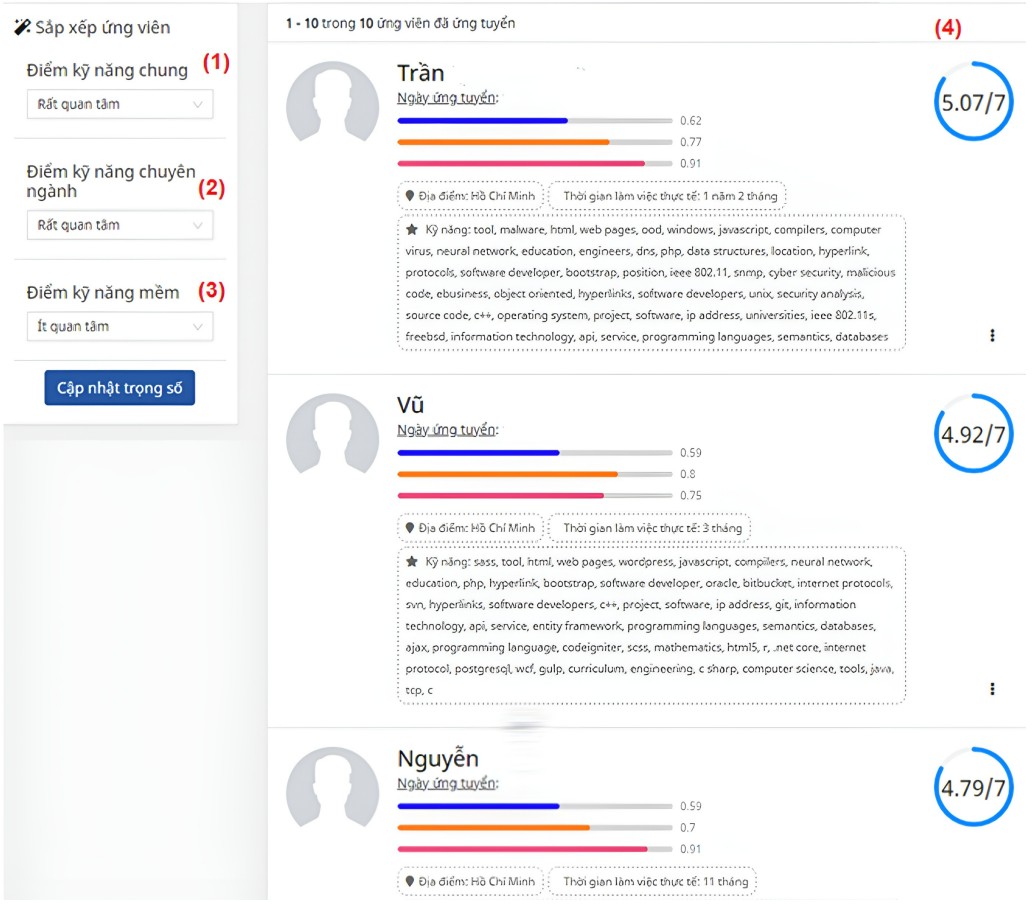

**Figure 9** **The page of the list of ranked CVs for the JD of a front-end engineer.** (1) General score (2) Domain score (3) Soft skill score (4) List of potential CVs.

**Table 4** **Results of matching front-end resumes to job description for front-end engineers.**

| ID | Domain score | General score | Soft skill score | Overall score |
|---|---|---|---|---|
| CV_2 | 0.77 | 0.62 | 0.91 | 5.07 |
| CV_8 | 0.80 | 0.59 | 0.75 | 4.93 |
| CV_1 | 0.70 | 0.59 | 0.91 | 4.79 |
| CV_6 | 0.69 | 0.58 | 0.90 | 4.78 |
| CV_9 | 0.73 | 0.59 | 0.55 | 4.53 |
| CV_5 | 0.66 | 0.58 | 0.70 | 4.42 |
| CV_0 | 0.62 | 0.51 | 0.64 | 4.02 |
| CV_3 | 0.63 | 0.51 | 0.55 | 3.96 |
| CV_4 | 0.57 | 0.52 | 0.55 | 3.82 |
| CV_7 | 0.51 | 0.51 | 0.55 | 3.59 |

Nguyen et al. (2024), *PeerJ Comput. Sci.*, DOI 10.7717/peerj-cs.1786

**Table 5** **The results of matching proportions of each experimented method with the list built by experts in cases.** The bold indicates the best values.

| Job description | Top 8 | | | | Top 5 | | | | Top 3 | | | |
|---|---|---|---|---|---|---|---|---|---|---|---|---|
| | Jaro–Winkler distance | Cosine | Levenshtein distance | Our method | Jaro–Winkler distance | Cosine | Levenshtein distance | Our method | Jaro–Winkler distance | Cosine | Levenshtein distance | Our method |
| JD_1 | 0.84 | **0.88** | **0.88** | **0.88** | **0.61** | 0.40 | 0.60 | 0.60 | 0.92 | 0.0 | 0.33 | **1.0** |
| JD_2 | **0.80** | 0.75 | 0.75 | 0.75 | 0.78 | 0.40 | 0.40 | **0.80** | 0.63 | 0.0 | 0.0 | **0.67** |
| JD_3 | 0.60 | 0.88 | 0.88 | **1.0** | 0.57 | 0.40 | 0.40 | **0.60** | 0.59 | 0.33 | 0.0 | **0.67** |
| JD_4 | 0.85 | **1.0** | 0.88 | **1.0** | 0.75 | 0.60 | **0.80** | **0.80** | 0.66 | 0.33 | 0.33 | **0.67** |
| JD_5 | 0.88 | 0.88 | 0.75 | **1.0** | 0.92 | 0.60 | 0.40 | **1.0** | 0.66 | 0.33 | 0.33 | **0.67** |
| JD_6 | 0.75 | 0.75 | 0.75 | **0.88** | **0.92** | 0.20 | 0.20 | 0.80 | 0.89 | 0.0 | 0.0 | **1.0** |
| JD_7 | 0.80 | 0.75 | **0.88** | 0.75 | 0.75 | 0.60 | 0.40 | **0.80** | 0.94 | 0.33 | 0.67 | **1.0** |
| JD_8 | 0.85 | **0.88** | 0.75 | **0.88** | **0.82** | 0.60 | 0.40 | 0.80 | 0.65 | 0.33 | 0.33 | **0.67** |
| JD_9 | 0.78 | **0.88** | **0.88** | 0.75 | 0.76 | 0.40 | 0.20 | **0.80** | 0.63 | 0.33 | 0.0 | **0.67** |
| JD_10 | 0.80 | 0.75 | **1.0** | 0.88 | 0.78 | 0.60 | 0.60 | **0.80** | 0.86 | 0.67 | 0.67 | **1.0** |

The experiment results indicate that the proposed method exhibits better-matching proportions than other methods in most cases. Particularly in real-world recruitment scenarios, recruiters typically shortlist only three to five potential candidates for a given job opening. Moreover, the proposed method demonstrates its superior performance when the Top 3 or Top 5 candidates are being evaluated.

## Experiment on the search system of labor laws

There are 17 chapters and 220 articles in *Vietnam's Labor Code 2019* (*Vietnam National Assembly, 2019*), while the *Law on Employment 2013* has seven chapters and 62 articles (*Vietnam National Assembly, 2013*). They are general rules that apply to all labor-related activities. To further structure the legal knowledge base from *Decree on Detailing Unemployment Insurance 2015* (*Government of Vietnam, 2015*), which contains the comprehensive unemployment insurance regulations, is also gathered. First, a database that follows the format Chapter–Section–Articles–Paragraph–Point is used to organize the legal documents. The substance and meaning of several terms in the labor law documents gathered from *Government of Vietnam (2020)* are then represented using the ontology Legal-Onto.

A labor law search system aims to retrieve labor law knowledge for user needs. This system is able to provide relevant information taken from legal documents when users request definitions for terms in labor legislation or any administrative processes which are linked to their work. In reality, the law search system concentrates on five categories of questions to suit the needs of individuals looking for IT jobs:

- Kind 1: Employment support policy
- Kind 2: Labor market information
- Kind 3: Issuing a national vocational certificate
- Kind 4: Organization and activities of employment services
- Kind 5: Unemployment insurance

First, the user enters a query into the search system. The system classified input questions into definitions, procedures, and/or related knowledge. After that, it returns responses to the users. When receiving results, the user will evaluate their usefulness for himself or herself. In addition, the results are also saved, and they will be examined by a lawyer and a law instructor in labor resources later.

**Example 5.1** The input query $q1 = $ "*What is unemployment insurance?*" The system will extract keywords from query $q$: "*What is,*" "*unemployment insurance*". The word "*what is*" is used to classify the query into kinds, declaring the meaning of a concept. The keyword "*unemployment insurance*" helps to find the concept. Then, from that point, the system returns the result for query q1:

"*According to Paragraph 3, Article 3, Law of Employment 2013:*

*3. Unemployment insurance is a regime to meet a part of an employee's income when he or she loses a job, to support workers in vocational training, job maintenance, and job search on the basis of contributions to the Unemployment Insurance Fund*".

**Table 6  Result of querying on the knowledge content of Vietnamese labor laws.**

| Kind | Queries | Number of correct results | | | | Proportions |
|---|---|---|---|---|---|---|
| | | Definitions | Procedure | Related knowledge | Total | |
| 1 | 42 | 11 | 9 | 6 | 26 | 62% |
| 2 | 48 | 10 | 10 | 7 | 27 | 56% |
| 3 | 59 | 19 | 15 | 5 | 39 | 66% |
| 4 | 36 | 9 | 8 | 5 | 22 | 61% |
| 5 | 24 | 5 | 7 | 4 | 16 | 67% |
| Total | 209 | 54 | 49 | 27 | 130 | 62% |

Table 6 shows the test results on the content about definition queries, asking about an administrative procedure, and related knowledge. The average accuracy of the search is more than 60%. The search can answer queries about definitions correctly. In addition, by using the ontology Legal-Onto to represent relations between knowledge in the law, the system can extract well-related knowledge. However, the system is not good with questions about administrative procedures. To complete an administrative procedure, there are many decisions from the government and instruction decrees from related ministries; thus, the identification of documents for the procedure is not accurate.

Figure 10 visualizes the results of 209 queries based on five kinds of queries. The system is not good when processing queries about labor market information because the data of this market are not collected fully. The system is most effective with queries about unemployment insurance (more than 70% accuracy). This is also an important requirement for candidates who are using the system to seek jobs.

Vietnamese Law (https://thuvienphapluat.vn/) is a library of legal documents in Vietnam. This program includes almost all documents in many majors. It is one of the largest law e-libraries in Vietnam. It can search for stored documents and their update over time.

Better Work is a special program that is held by the International Labor Organization (ILO) and the International Finance Corporation (IFC). It released an application of Vietnam Labor Law Guide (Vietnam LLG; https://play.google.com/store/apps/details?id=org.betterwork.llg&pcampaignid=web_share) to strengthen compliance with labor standards and enhance the capabilities of businesses in the global apparel industry (Fig. 11). This application can run on the Android and iOS operating systems. It covers the content of Vietnamese Labor Laws, provisions on occupational safety and health, and minimum wages. It also includes frequently asked questions (FAQs) and quiz questions.

Because Vietnamese Law and Vietnam LLG only support the search for legal documents, Table 7 compares them and our system when finding legal documents about the abilities to store documents and search for the laws.

## Comparing resume and labor law search systems

Currently, many systems support the search for resumes and legal documents. For example, Gambaru (https://gambaru.io/en/search-matching), TopCV (https://www.topcv.vn/) and Indeed (https://vn.indeed.com/?r=us) are useful systems for job seeking in the IT

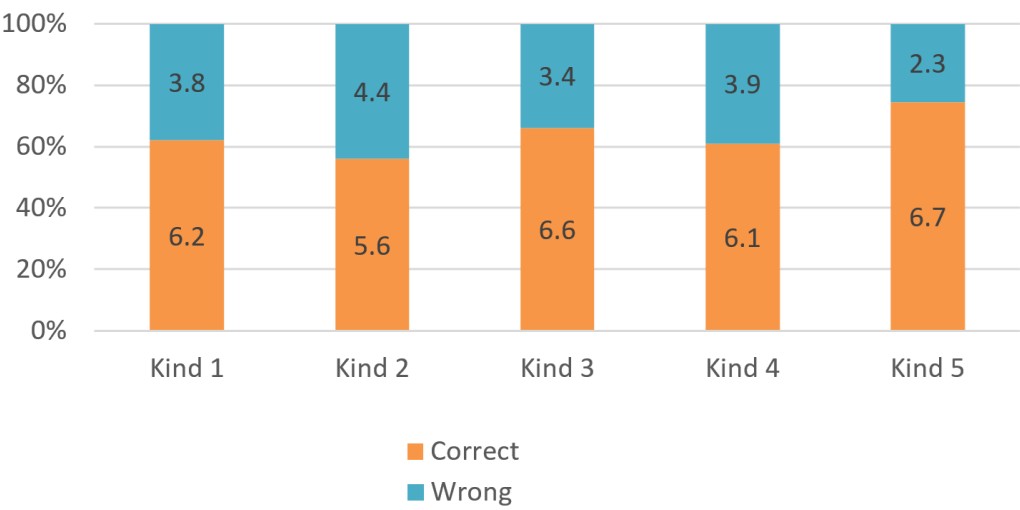

**Figure 10** The precision of the querying system on each content of Vietnamese labor laws.

domain, and Vietnamese Law (https://thuvienphapluat.vn/) and the National Database (http://vbpl.vn/botuphap/Pages/Home.aspx) about Law Documents support finding legal documents. However, those systems have not yet completely combined these functions to become convenient for users. Those functions are separate; they are not connected to work, and they also have several limitations to meet practical requirements.

Gambaru is a recruitment platform with many experienced IT sources. This system also supports talent searching for jobs that are suitable for their skills. It helps to place the high calibers for the goal of the job through AI technology. TopCV is a website that helps users create their CVs and seek suitable jobs with candidate majors. It also helps entrepreneurs search for candidates to recruit. Table 8 compares Gambaru, TopCV, and our system as a combined system of resume search and support for labor law search.

## CONCLUSION AND FUTURE WORK

This article describes the structure of the ontology Job-Onto. In the IT domain, this ontology effectively represents CVs and JDs. Job-Onto is utilized to develop a system for job seekers to search for resumes. In addition, many problems associated with matching CVs and JDs are proposed and resolved using data collection, skill extraction, and skill graph generation. The designed system is capable of extracting skills from CVs and JDs and evaluating their compatibility. The system can provide the highest-ranking positions (or candidates) for a given CV (or JD).

In addition, the structure of Legal-Onto, an ontology representing the domain of labor law knowledge, is proposed. Legal-Onto combines the ontology Rela-model (*Do, Nguyen & Selamat, 2018*) and a conceptual graph's structure. On the basis of the Legal-Onto ontology, a number of issues pertaining to the search for labor laws are investigated, including the legal policies governing unemployment and healthcare benefits. Using its knowledge of

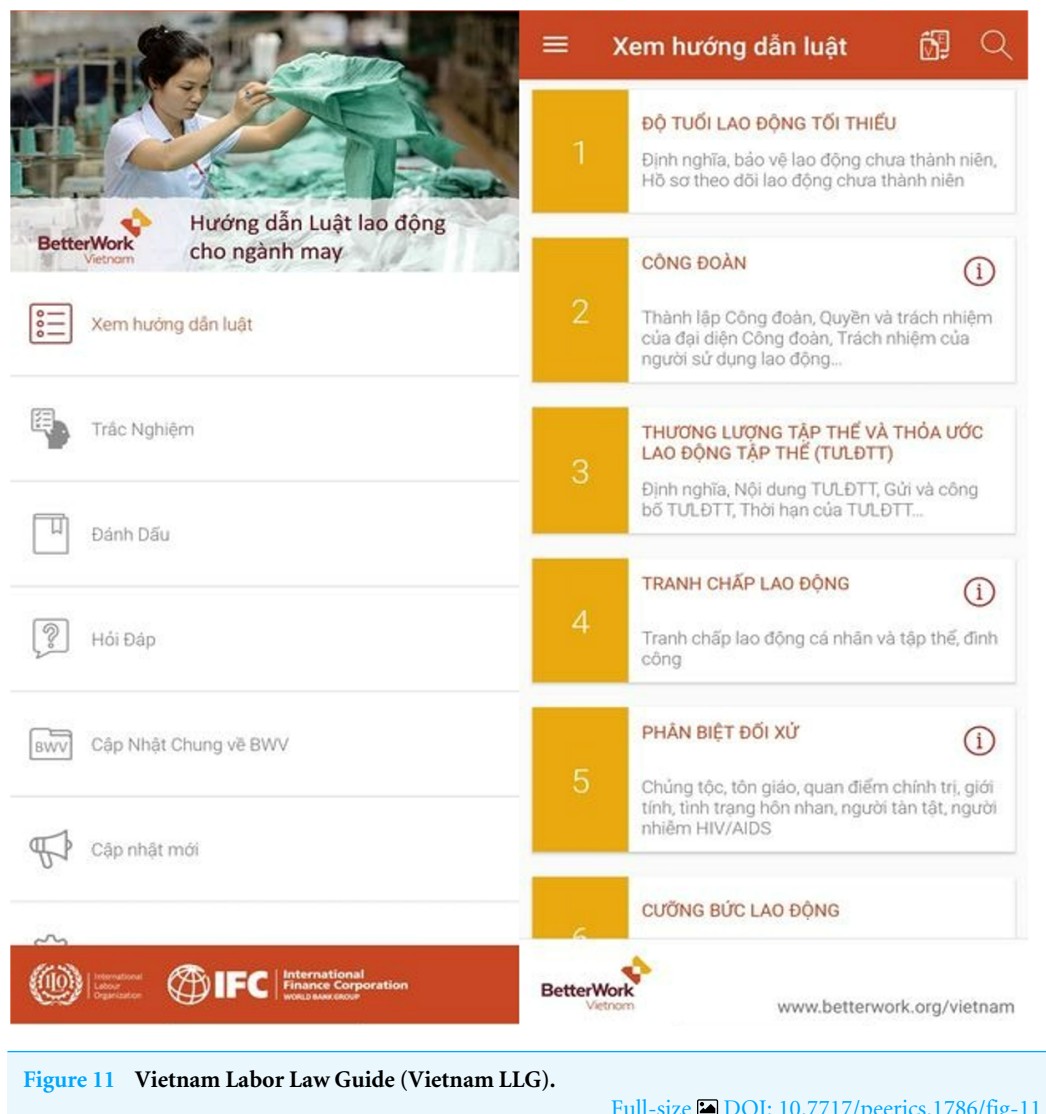

**Figure 11** **Vietnam Labor Law Guide (Vietnam LLG).**

labor law documents, the search system can retrieve the law's content in response to an input query.

On the basis of the ontology integrating Job-Onto and Legal-Onto, the architecture of an intelligent resume-search system in the IT domain is subsequently constructed. This system assists users in searching for employment opportunities and Vietnamese labor documents. The experimental findings demonstrate that the designed system is effective for resuming the search for unemployment and healthcare benefit policies. This study has designed a useful method to integrate two ontologies for constructing a search system combining seeking IT jobs and searching labor law. This system is useful for unemployed individuals seeking employment through the system and understanding insurance policies for the unemployed to ensure their benefits from the government.

To expand application domains in the future, several automatic methods to enrich ontology will be studied for data collection. The enhancement techniques for extracting

**Table 7 Comparison of the Vietnamese Law, Vietnam LLG, and our system in searching for legal documents.**

| | Vietnamese law | Vietnam LLG | Our system |
|---|---|---|---|
| | Those systems can store all documents with their structures, including Chapter, Section, Article, Clause, and Point. | | |
| Ability for storing legal documents | ●Store almost legal documents in many majors.<br>● Only store the content of law documents and note several changes on it over time. | ●Store Vietnamese Labor Law and provisions on occupational safety and health and minimum wages.<br>● Only store the content of law documents. | ● Store Vietnamese Labor Law, Law on Employment, and Law on Social Insurance.<br>● The representation is worked as an intellectual model representing relations between these legal documents and their semantics. |
| Ability for law search | The search is worked on the database of law documents:<br>● It can show the content of a law document.<br>● It cannot search based on a query sentence, thus, it does not give an answer to a question. | Searching by keywords:<br>● It cannot perform a semantic search with multiple keywords.<br>● It only returns all documents that contain search keywords. | The system supports searching the knowledge of legal documents:<br>● The system can give an answer for the meaning of an input query.<br>● The semantics between input keywords is represented by a knowledge graph, and suitable information from the knowledge base is extracted. |

information from a CV will be useful in the real world as a result of the extremely large amount of data collected. In addition, there is a great deal of room for the development of a method for representing legal knowledge using multiple documents. To be more effective in legal consulting, the knowledge model can be organized by ontology and graph embedding approach (*Arnold & Rahm, 2014*; *Ngo, Nguyen & Le-Khac, 2023*). The graph will help to represent relations between entities in legal documents more accurately. Moreover, this legal searching function will also be implemented as a question-answering system for residents (*Kourtin, Mbarki & Mouloudi, 2021*; *Dang et al., 2023*). This makes it more convenient to use in practice.

## Funding
This research is supported by Vietnam National University, Ho Chi Minh City (VNU-HCM) under grant number DS2023-26-04. The funders had no role in study design, data collection and analysis, decision to publish, or preparation of the manuscript.

## Grant Disclosures
The following grant information was disclosed by the authors:
Vietnam National University, Ho Chi Minh City (VNU-HCM): DS2023-26-04.

## Competing Interests
The authors declare there are no competing interests.

## Author Contributions
● Hien Nguyen conceived and designed the experiments, analyzed the data, authored or reviewed drafts of the article, and approved the final draft.

**Table 8 Comparison of Gambaru, TopCV, and our system in terms of the criteria for the software development and functions of a combined system in resume and labor law searching.**

| | Gambaru | TopCV | Our system |
|---|---|---|---|
| **About the software development** | | | |
| Operability | • This system is an online platform. The information is shown clearly and sightly. <br> • The user-interface is friendly and convenient for users. | • This program works as a web-site that users can access easily. <br> • The user interface is friendly, easy to use, especially for the purpose of resume searching. | • The system works online. The system is divided into two main functions: Resume searching and searching for labor law as a re-trieval system, especially for queries about unemployment insurance. <br> • The user interface is hard; it is not smooth. The design of the system is simple; thus, users can use it easily. |
| | • Users can update the information about their CVs of JDs. <br> • The law is able to be updated by the administrator. | | |
| Flexibility | • The system gives results based on the job title of an input JD (or CV). <br> • The system only allows users to read the content of several articles about labor laws. | • The system gives results based on the job title of an input JD (or CV). <br> • The system only allows users to read the content of several articles about labor laws. | • The system explains the importance of the knowledge material. As a result, consumers require the extracted knowledge. <br> • The system allows users to find the answers to complex questions with more than two patterns. |
| **About functions of systems** | | | |
| | • The system can recommend candidates (or jobs) for an input job description (or a CV). | | |
| Resume searching | • The system can evaluate the matching between skills from a CV and a JD. This matching used an ontology to build the system of skills. | • The ranking of the system is based on job positions and some of their specific skills. It has not yet given suitable results for input informa-tion. | • The ranking of the system is worked based on skills ontology, such as domain skills, soft skills, and general skills. |
| Searching for labor laws | • It is not a main function of this system. It only stores some contents of Labor on Employment. | • This website only supports show-ing the content of several deter-mined articles about labor laws. <br> • The system cannot extract infor-mation for an input query. | • The system equipped a knowledge base about labor laws; thus, it can simulate a part of a consultancy system in law. <br> • The system can retrieve the content of the law for an input query, especially to become more effective with queries about unemployment insurance. |

- Vuong Pham conceived and designed the experiments, analyzed the data, authored or reviewed drafts of the article, and approved the final draft.
- Hung Q. Ngo analyzed the data, authored or reviewed drafts of the article, and approved the final draft.
- Anh Huynh performed the experiments, performed the computation work, prepared figures and/or tables, authored or reviewed drafts of the article, and approved the final draft.
- Binh Nguyen performed the experiments, prepared figures and/or tables, authored or reviewed drafts of the article, and approved the final draft.
- José Machado conceived and designed the experiments, analyzed the data, authored or reviewed drafts of the article, and approved the final draft.

## Data Availability

The data and code are available at figshare: Nguyen, Hien (2023). Dataset for Intelligent Searching System on Resumes and Legal documents. figshare. Dataset. https://doi.org/10.6084/m9.figshare.24818709.v1.

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
