# Peer review of "Intelligent search system for resume and labor law"

_PeerJ Computer Science, doi:10.7717/peerj-cs.1786_

## Round 0.1 · original submission · Major Revisions

Please consider the reviewers comments.

**Language Note:** PeerJ staff have identified that the English language needs to be improved. When you prepare your next revision, please either (i) have a colleague who is proficient in English and familiar with the subject matter review your manuscript, or (ii) contact a professional editing service to review your manuscript. PeerJ can provide language editing services - you can contact us at copyediting@peerj.com for pricing (be sure to provide your manuscript number and title). – PeerJ Staff

Reviewer 1 ·

Basic reporting

Overall it is clear and unambiguous.
In abstract "The demand for online job searching and searching for labor regulations in legal documents, particularly regarding the policy for unemployment beneûts, is essential." Given the problem is essential, what is the painpoint of the current status? Why the proposed method can solve such painpoint? It would be great if authors can further polish the motivation in abstract.
Also it is a bit unclear the definition of 'ontology'. It would be helpful to discuss what is 'ontology' in the problem domain and why it is important.
page 4 line 177 "The Job-Onto ontology" I guess there should be no space?

Experimental design

Overall I think this paper lacks of strong benchmark.

Validity of the findings

Overall I think this paper lacks of strong benchmark. Also I am not sure if the current model has significant improvement over the very simple benchmarks (Cosine, Levenshtein).
I would recommend authors to repeat the experiment to understand the confidence interval of the metrics, as well as the statistical significance of the difference.

Additional comments

My major concern is the focus of this paper. It seems the paper build a new system, and its innovation is mostly focus on its problem space (job search). It lacks of methodology innovation to general audience in the computer science domain and thus may not a good fit for this journal. I feel it may better fit into labor market related journals.

Cite this review as
Anonymous Reviewer (2024) Peer Review #1 of "Intelligent search system for resume and labor law (v0.1)". PeerJ Computer Science

Reviewer 2 ·

Basic reporting

No comment!

Experimental design

No comment

Validity of the findings

No comment

Additional comments

It should be careful with the acronymous.
Definition after they are used or without definitions.

Cite this review as
Anonymous Reviewer (2024) Peer Review #2 of "Intelligent search system for resume and labor law (v0.1)". PeerJ Computer Science

·

Basic reporting

English language used in the text is clear and professional.

Introduction and background clearly show the context and the objective of the text. The literature is relevant and it is correctly referenced in the text.

Structure of the text conforms the PeerJ standards.

For Fig. 1, 5, 6, 7, 8 I would suggest resize and restructure so all the inscriptions would be readable. Fig. 9 is too blurry and it cannot be read at all, so it will bring no information to the reader. Maybe the latter screenshot can be omitted in the publication? Fig. 10 also can be resized a bit.

Tables are clear, and they explain the results that are discussed in the text.

Raw data. The implementation of the software and data used in experiments is provided by the authors. Implementation is composed by a backend and a frontend systems. The code looks well structured and organized.

Experimental design

Originality of the research. Job description ontologies are known technique in the specialized literature. It is not clear from the proposed text what exactly makes the proposed method unique? Where exactly is the originality of the proposed solution? Is there a scientific novelty, or the text describes a working implementation of existing technologies.

Definition of the research questions is clear.

Investigation performed. The objective of the performed experiments can be stated more clearly in the text.

Detail of the methods description. The authors presented well structured software implementation in raw data. The paper will gain clarity, if the authors describe in more details the algorithms which are implemented by their software. The must point out clearly which of the algorithms and methods they provide bring originality of their work.

Validity of the findings

Impact and novelty can be described better if authors describe which algorithms and methods are their work.

Additional comments

I think there is a serious research that is behind the proposed text. However, the adoption of ontologies for these kind of applications is not something new. The authors must clearly show what is the original part of their proposition. The relation to the COVID-19 sounds more like a redundant cliche and in my opinion the text is not going to loos quality if it is omitted. There are mathematical-like expressions in the text, that are not completely clear: better, more conservative notations can be used, or at least be careful if you have neighboring letters like in the case of
--> t f(p, d)
how should we interpret t here?
tf -- one term
t multiplied by f(p, d) ?
Please, follow the formal rules for mathematical expression composition.

Cite this review as

---

## Round 0.2 · Minor Revisions

Please consider the final very minor comments from R3.

·

Basic reporting

### English language

English language used in the manuscript is clear and on a professional level.

### Intro, background and references

The abstract and the introduction clearly state the objective of the manuscript. The introduction contains a comprehensive literature survey on the topics of machine learning techniques, including NLP methods with applications to job/CV matching. The introduction clearly states the location of the proposed ontology-based approach regarding the existing methods in the literature.

### Text structure

The structure of the text is with respect the journal standards.

### Figures and tables

Figures and tables are clear and with good structure. Still I would recommend to enlarge the font in the figures where it is possible (especially in Figures 5 and 6).

### Raw data

The source code code of the implementation is well structured and organized.

Experimental design

### Originality of the research and scope of the journal

The research is original and it is within the scope of the journal.

### Research questions definition

Research questions are well and clearly defined by authors.

### Technical and ethical standards of the research

The manuscript contains a clear system of definitions and also clearly shows the structure of the proposed solution. The experiment of the authors is based on the source code that is provided by the authors. The manuscript also contains a pseudocode that clearly shows the algorithm that finds matched knowledge of a given query.

### Description detail sufficiency to replicate

The details of the description is sufficient.

Validity of the findings

### Impact and novelty

The research is novel and it proposes an improvement of the ontology-based approach in the particular practical application for job matching with applicants CVs.

### Data provided robustness and statistical control

The experiments described in the manuscript are statistically meaningful.

### Conclusions

The section Conclusion and Future Work clearly summarizes the results of the research.

Additional comments

Please, improve the font in Figures 5 and 6.

Cite this review as

---

## Round 0.3 · accepted · Accept

The authors addressed the issues raised in the previous comments. It can be accepted.